# Long-Range Static and Dynamic Previtreous Effects in Supercooled Squalene—Impact of Strong Electric Field

**DOI:** 10.3390/molecules26195811

**Published:** 2021-09-25

**Authors:** Szymon Starzonek, Aleksandra Drozd-Rzoska, Sylwester J. Rzoska

**Affiliations:** Institute of High Pressure Physics of the Polish Academy of Sciences, 29/37 Sokołowska Street, 01-142 Warsaw, Poland; arzoska@unipress.waw.pl (A.D.-R.); sylwester.rzoska@gmail.com (S.J.R.)

**Keywords:** molecular liquid, nonlinear dielectric effect, glass transition, ODIC, pretransitional effect

## Abstract

This article presents evidence for the long-range previtreous changes of two static properties: the dielectric constant (*ε*) and its strong electric field related counterpart, the nonlinear dielectric effect (NDE). Important evidence is provided for the functional characterizations of *ε*(*T*) temperature changes by the ‘Mossotti Catastrophe’ formula, as well as for the NDE vs. *T* evolution by the relations resembling those developed for critical liquids. The analysis of the dynamic properties, based on the activation energy index, excluded the Vogel–Fulcher–Tammann (VFT) relation as a validated tool for portraying the evolution of the primary relaxation time. This result questions the commonly applied ‘Stickel operator’ routine as the most reliable tool for determining the dynamic crossover temperature. In particular, the strong electric field radically affects the distribution of the relaxation times, the form of the evolution of the primary relaxation time, and the fragility. The results obtained in this paper support the concept of a possible semi-continuous phase transition hidden below *T_g_*. The studies were carried out in supercooled squalene, a material with an extremely low electric conductivity, a strongly elongated molecule, and which is vitally important for biology and medicine related issues.

## 1. Introduction

The problem of supercooled liquids has been an intensive field of research for decades. Nevertheless, it remains one of the most fundamental challenges for 21st-century science. One of the hallmark features that attracts particular attention are the long-range previtreous changes of dynamic properties, starting even at 100 K above the glass temperature *T_g_*, for viscosity *η* or for the equivalent primary relaxation time *τ* There is a widespread belief that previtreous effects are related only to dynamic properties, which exhibit some of the hallmarks of universality and are conceptually expressed by the Super-Arrhenius (SA) relation:(1)$$\tau \left(T\right)=\tau_{0}exp\left(\frac{E_{a}\left(T\right)}{RT}\right)$$
where $E_{a}\left(T\right)$ denotes the apparent and temperature-dependent activation energy. Parallel relations take place for η(T) or σ(T) behaviour. It is empirically stated that the glass temperature *T_g_* may be defined as a temperature when τ(T)≈100s and $\eta \left(T\right)\approx 10^{13}Poise$. For a low molecular weight liquid, the pre-exponential factor is $\tau_{0}\sim 10^{-14}s$. For the basic Arrhenius dependence $E_{a}\left(T\right)=E_{a}=const$ in the given temperature domain.

There are some indications for previtreous changes in the heat capacity, which is the most representative thermodynamic magnitude. However, these changes only appear in a narrow range of temperatures above *T_g_*, and to date, no common functional description has been found. 

This article presents evidence for the long-range previtreous changes of the dielectric constant, which is the basic ‘static’ property. These changes are associated with the specific previtreous evolution of dynamic properties. Moreover, these changes extend to the behaviours under the strong electric field, which has been tested in a way that has been, so far, relatively unexplored.

The aforementioned behaviours have been found in squalene ((C_5_H_8_)_6_, triterpene), a compound with enormous biological [1] and medical potential [2,3], that has been used in the pharmaceutical [4,5] and food [6,7] industries due to its emollient, skin hydration, antioxidant, and anticancer properties. The dietary presence of squalene in olive oil was suggested as the key factor in reducing cancer mortality in Mediterranean countries [7]. However, both its synthesis and characterization still constitute a challenge for chemists and biochemists [8,9,10]. From this point of view, squalene, i.e., 2,6,10,15,19,23-hexamethyl-2,6,10,14,18,22-tetracosahexaene, is a naturally occurring terpenoid hydrocarbon with a strongly elongated form and a weak permanent dipole moment (μ≈0.6D) [11]. The important feature of squalene is its extremely low electric conductivity, which facilitates measurements under the strong electric field and enables the minimization of the biasing effect impacts of the electric conductivity.

When discussing the previtreous dynamics, it is notable that the SA Equation (1) mainly has a cognitive meaning and cannot be used for portraying the experimental data due to the unknown form of the apparent activation energy. Consequently, a replacement scaling relation must be developed. For decades the leading position was the Vogel–Fulcher–Tamman (VFT) relation:(2)$$\tau \left(T\right)=\tau_{0}exp\left(\frac{D_{T}T_{0}}{T-T_{0}}\right)$$
where $T_{0}$ is the VFT estimation of the ideal glass temperature, which is also linked to a hypothetical phase transition; $D_{T}$ is the fragility strength coefficient.

Comparing Equations (1) and (2), one obtains: $E_{a}\left(T\right)=RD_{T}T_{0}/t$, where $t=\left(T-T_{0}\right)/T$ is the relative dimensionless distance from the singular temperature $T_{0}$. It is useful to recall the link of the SA Equation (1) to the fragility, or the basic ‘quasi-universal’ metric for the previtreous behaviour, which shows the distortion from the reference Arrhenius behaviour:(3)$$m={\left[\frac{d{log}_{10}\tau \left(T\right)}{d\left(T_{g}/T\right)}\right]}_{T=T_{g}}$$

For the basic Arrhenius behaviour, $m={log}_{10}\tau \left(T_{g}\right)-{log}_{10}\tau_{0}\approx 16$. The weak slowing down below the Arrhenius pattern is related to m<30 and is referred to as the ‘strong’ glass former. Systems with the essential SA dynamics, and m>30, are called ‘fragile’. The maximal reported value is: m>200 [12,13,14,15]. One may extend Equation (3) into the previtreous domain by introducing $m\left(T>T_{g}\right)$ as the apparent fragility, which is also known as the steepness index. By comparing Equations (1)–(3), one obtains: $D_{T}=590/\left(m-16\right)$. The enormous previtreous slowing down indicates the broadband dielectric spectroscopy (BDS) as the key research tool because it can cover 12–17 decades in frequency/time during a single measurement process. The BDS can detect the primary relaxation time, as well as the distribution, the other relaxation processes, the DC electric conductivity, etc. [16,17,18,19,20]

In the last decade, the fundamental authority and importance of the VFT relation have been questioned. This, in turn, has led to the emergence of competing dependences. To this end, it is worth recalling the MYEGA equation, which avoids the final temperature singularity [21]:(4)$$\tau \left(T\right)=\tau_{0}exp\left(\frac{K}{T}exp\left(\frac{C}{T}\right)\right)$$
where $T>T_{g}$, and *K* and *C* are constants.

For liquid-crystalline and plastic crystal glass formers, the prevalence of the critical-like description was shown as [22,23,24,25,26]:(5)$$\tau \left(T\right)=\tau_{0}{\left(T-T_{C}\right)}^{-\varphi }$$
where $T>T_{g}$ and $T_{C}<T_{g}$; while the power exponent 8.5<φ<15.

Such basic properties as the dielectric constant, primary relaxation time, or electric conductivity are precisely determined from broadband dielectric spectroscopy (BDS) scans. The maturity of the BDS method that is applied for glass-forming systems implies the question of its successor. The natural candidate is the Nonlinear Dielectric Spectroscopy (NDS), which is the strong electric field related to BDS [27,28,29,30,31,32,33,34,35,36,37]. By considering the polarizability as the function of the intensity of the electric field, one obtains [37]:(6)$$P\left(E\right)=\chi^{*}E+\Delta \chi^{*}E^{3}+\ldots$$
where $\chi^{*}=\varepsilon^{*}-1$ is for the complex dielectric susceptibility, and $\varepsilon^{*}$ is for the dielectric permittivity.

It is well known that BDS is directly coupled to the 2-point correlation function, whereas NDS is coupled to the 4-point correlation function. This fact shows that an NDS-based study can significantly expand insight into the unknown nature of the previtreous state. The progress made in NDS studies of glass formers has led mainly to developing the phenomenological description of ‘nonlinear’ spectra in various materials. However, to date, no previtreous behaviour under the strong electric field has been postulated [29,30,31,32,33,34].

Studies of pretransitional phenomena have appeared to be key for the grand success of the Physics of Critical Phenomena [38,39,40]. However, there are striking differences between the precritical and the previtreous phenomena, namely: (i) the precritical phenomena are associated with a well-defined phase transition; (ii) the practical changes are observed for dynamic, static, and thermodynamic physical properties; and (iii) the universality is associated not only with the form but also the values of universal parameters (i.e., critical exponents, and the ratios of critical amplitudes) [38,39,40,41,42,43,44,45]. These basic features seem to be absent, or at least disputable, for the glass transition and the previtreous effects [12,13,14,15,16,17,27,28,29,30,31,32,33,34,35,36,37].

This article provides evidence for the long-range previtreous behaviour of the dielectric constant and its strong electric field-related counterpart, the nonlinear dielectric effect, or the NDE in supercooled squalene. The dielectric constant follows the ‘Mossotti Catastrophe’ functional pattern, which has so far been considered the ‘forbidden state’ in dipolar liquids. This result is supplemented by analysing the primary relaxation time evolution under the weak and strong electric fields. The application of the distortions-sensitive analysis has revealed features of the previtreous dynamics. This analysis minimises the central problem of data analysis in the previtreous region, which is due to its location well above the hypothetical, hidden, and singular temperatures. 

## 2. Results and Discussion

This article is focused on the behaviour under strong and weak electric fields, including the dynamic and static issues related to dielectric studies. However, before the basic reference, the BDS-related behaviour in supercooled squalene is presented. Figure 1a shows selected spectra for the imaginary part of the dielectric permittivity in supercooled squalene in the vicinity of the glass temperature. The peaks of main loss curves determine the primary relaxation times $\tau =1/\omega_{peak}=1/2\pi f_{peak}$ [15,27]. In the immediate vicinity of the glass temperature, estimated via the empirical condition $\tau \left(T_{g}\right)\approx 100s$ as $T_{g}\approx 164\;K,$ the secondary fast relaxation (beta) process emerges. For slightly higher temperatures, the low-frequency relaxation (LF) process, disappearing on $T\to T_{g}$, was observed. The temperature evolutions of basic relaxation processes resulting from the analyses of dielectric spectra are shown in Figure 1b. The primary relaxation time shows clear SA behaviour (Equation (1)). Its optimal portrayal is further described in this article. Well below *T_g_*, the low-frequency process (LF) also occurs, and in the immediate vicinity of *T_g_*, the secondary (beta) process emerges and smoothly continues deeper into the solid glass state, for $T<T_{g}$.

Stevenson and Wolynes [46] have suggested the universal origins of the beta process by showing that by adding fluctuations to the existing structure of the random first-order transition theory, a tail develops on the low free energy side of the activation barrier distribution, which shares many of the observed features of the secondary relaxations. Consequently, they have suggested that while primary relaxation takes place through activated events involving compact regions, secondary relaxation is governed by more ramified, string-like, or percolation-like clusters of particles. When considering Figure 1b, one may conclude that such a mechanism must be valid also in the solid glass states below *T_g_*_,_ where bonds stabilizing the liquid-like arrangement in the amorphous solid form develops.

The analysis of the SA temperature evolution for the primary relaxation time most often involves its portrayal by the selected model dependence (e.g., Equations (3)–(5)) by the determination of the fragility (Equation (2)) and the search for the dynamical crossover phenomenon via the so-called Stickel analysis based on the plot ${\left[dln\tau \left(T\right)/d\left(1/T\right)\right]}^{-1/2}$ vs. 1/T [15,47]. In this article, we avoid the latter because it is assumed a priori that the VFT in Equation (3) obeys [48,49]. Prior to fitting experimental data by an arbitrarily selected model-relation, the optimal model equation was found. Finally, instead of focusing on the value of the fragility at *T_g_* (Equation (2)), the analysis of the apparent fragility and the activation energy for the whole previtreous domain was performed. These procedures are implemented below, and the behaviour under both weak and strong electric fields is discussed.

The survey of the available results concerning the previtreous domain revealed a surprising gap regarding the temperature behaviour of such basic properties as the dielectric constant [12,13,14,15,16,17,21,22,23,24,25,26,27,28,29,30,31,32,33,34,35,36,37] and refs. therein. Our results for squalene eliminate this gap (see Figure 2). The presented value is always related to the midpoint of the static domain and considers its strong shift towards the lower frequencies on cooling towards *T_g_*. The lack of DC conductivity has an impact in the case of squalene. The scale applied in Figure 1 reveals that the temperature behaviour described in the broad temperature range also affects the obtained results by the relation:(7)$$1/\varepsilon =a+bT,\;\chi =\varepsilon -1=\frac{1}{a+bT}-1=\frac{\frac{1}{b}}{T-\frac{a}{b}}-1=\frac{A}{T-T^{+}}-1$$
(8)$$\mathrm{and}\;\mathrm{then}\;T^{+}=a/b\;\;\mathrm{and}\;A=1/b.$$

For the high-temperature domain $T_{B}\left(?\right)<T<T_{b}$, one obtains a=0.142 and $b=2.23\times 10^{-4}$, and for $T<T_{B}\left(?\right)\;a=0.181_{}$, and $b=4.43\times 10^{-5}$, where $T_{b}$ is a hypothetical secondary cross-over temperature.

The distortions-sensitive plot in the inset to Figure 2 confirms the validity of Equation (7) by additionally showing temperature $T_{B}$, for which the coincidence with the dynamic crossover temperature [15] can be considered.

Particularly important is the correlation of Equation (7) with the dependence known for the so-called ‘Mossotti Catastrophe’, appearing when applying the Mossotti-Lorentz (ML) local field for dipolar liquids. Its occurrence is explained as the consequence of neglecting the intermolecular interactions in the ML model. Its picturesque illustration is the ‘impossible ferroelectricity’ for the water appearing at *T* ~ 1100 K. Figure 2 shows the evidence for such behaviours in the supercooled squalene in the high temperature (*T > T_B_*) and low temperature (*T < T_B_*) dynamic domains. Only recently, a similar ‘Mossotti Catastrophe’ type behaviour was reported in a liquid and orientational disordered crystal (ODIC) phaser of glass-forming cyclo-octanol.

One of the targets of the presented studies was the behaviour of supercooled squalene under the strong electric field. Figure 3 focuses on dynamic issues by presenting the shift of primary loss curves under the strong electric field close to *T_g_*. The obtained behaviour fairly coincides with one known from reported NDS studies. Notwithstanding, some features characteristic for the given system are worth stressing. The first is the disappearance of the secondary relaxation process under the strong electric field. This means that structural relaxation-related differences between the hypothetical multimolecular species/clusters/strings and their surroundings disappear under the strong electric field. The second feature is related to the inset, which reveals differences in the distribution of the primary relaxation times, and recalls the concept from Jonsher [50] via relations:(9)$$\frac{d{log}_{10}\varepsilon \prime \prime \left(f\right)}{d{log}_{10}f}=m\;\;\;\;\;\;\mathrm{for}\;f<f_{peak}\;\;\;\mathrm{and}\;\;\;\frac{d{log}_{10}\varepsilon \prime \prime \left(f\right)}{d{log}_{10}f}=-n\;\;\;\;\;\;\mathrm{for}\;f>f_{peak}$$
where *m* and *n* denote the coefficients describing the distribution of the relaxation time for the low- and the high-frequency parts of the loss curve, respectively.

The distribution of the relaxation times is strongly non-Debye. For the spectrum under the weak electric field, m~1/2 (the low-frequency part) and n~1/2 (the high-frequency part). The latter was previously indicated by Olsen and Dyre [51], and later also in [52,53] as a possibly universal value for the $T\to T_{g}$ path. However, under the strong electric field, the high-frequency part of the primary relaxation loss curve broadens:(10)$$n\left(T\to T_{g}\right)\sim 1/3.\;$$

The example of the distribution of the relaxation processes related to the increment induced by the strong electric field is shown in Figure 4. It is noteworthy that its form is different from the one reported for the supercooled low molecular weight liquid and resembles the one observed in glass-forming orientationally disordered crystals. This scan suggests that for the ultraviscous supercooled squalene, the detected processes are mainly associated with the orientation of a permanent dipole moment. The hindering of translational molecular motions may be associated with the elongated and complex molecular structure of squalene.

Figure 5 enables the comparison of the temperature evolution of the primary relaxation times under the weak and strong electric fields. The inset presents the difference in the relaxation time under the weak and strong electric fields.

The reliable portrayal of the previtreous changes of the primary relaxation times remains a non-conclusive challenge despite decades of studies. This competition for scaling relations has yielded comparably within the limit of the experimental error and a new fitting quality has been developed. In fact, such experimental evidence creates an essential problem for any theoretical modelling. In the opinion of the authors, the non-conclusiveness of studies can be explained by the fact that the analysis is carried out above *T_g_*, i.e., well beyond a hypothetical singular temperature included in the experimental scaling relation. The experimental error strengthens the problem. As a possible solution for the problem, we can offer the distortions-sensitive analysis. We propose to apply to the experimental data presented in Figure 5 the analysis exploring the apparent activation index, based on the following transformation [54,55,56]:(11)$$\tau \left(T\right)\;\;\;\Rightarrow \;\;\;I_{DO}\left(T\right)=-\frac{dlnE_{a}\left(T\right)}{dlnT}=\frac{dE_{a}\left(T\right)/E_{a}\left(T\right)}{dT/T}$$

The application of Equation (11) requires a priori knowledge of the apparent activation energy. In [54,55], it was calculated in the previtreous domain using SA Equation (1), namely $E_{a}\left(T\right)=RTln\left(\tau /\tau_{0}\right)$, assuming the universal and constant value of the pre-factor $\tau_{0}=10^{-14}s$. However, such an analysis can yield strongly biased values. To avoid this, a new way of determining $E_{a}\left(T\right)$ was proposed in [56,57,58]. For a given set of τ(T) experimental data, a numerical solution was used. Based on the differential equation derived from the SA Equation (1):(12)$$\frac{H_{a}\left(T\right)}{RT}=\frac{dE_{a}\left(T\right)}{d\left(1/T\right)}+E_{a}\left(T\right)$$
where $H_{a}\left(T\right)=dln\tau \left(T\right)/d\left(1/T\right)$ denotes the apparent activation enthalpy.

The analysis is supplemented by the final numerical filtering cleaning. Based on these results, the activation energy index in the ultraviscous domain has been analysed. It was empirically found that in each case, the reciprocal of the index follows the simple linear dependence [56,57,58]: $\frac{1}{I_{DO}\left(T\right)}=aT+b$. It is worth stressing that such behaviour agrees with the indexes derived for the basic model equation, namely [56]: (i) $1/I_{DO}=\left(1/T_{0}\right)T-1$ for the VFT Equation (2); (ii) $1/I_{DO}=\left(1/C\right)T$ for the WM (MYEGA) Equation (3); and (iii) $1/I_{DO}=\left(1/\varphi \right)T-\left(T_{C}/\varphi \right)$ for the critical-like Equation (4). It was empirically proved that for systems studied in [56], the coefficient n=−1/b is located between 0.2 and 1.6. The highest value is for glass formers showing the uniaxial symmetry, such as liquid crystalline, where the critical-like behaviour is preferred (Equation (5)). Values *n* ~ 0.2 were noted for orientationally disordered crystals. Values n≈1 can be linked to the VFT Equation (2): in [56] it was called the ‘no-symmetry case’. Figure 6 shows the behaviour of the apparent activation index (A) determined for the ultraviscous supercooled squalene, as well as the configurational entropy *S_C_* (B), given by:(13)$$S_{C}\left(T\right)=S_{0}{\left(1-\left(\frac{b}{a}\right)\frac{1}{T}\right)}^{-1/b}=S_{0}{\left(1-\frac{T_{N}}{T}\right)}^{n}$$
where $T_{N}=\left|b/a\right|$ is the singular temperature denoted the Kauzmann temperature $T_{N}=T_{K}$, and n=−1/b describes the power exponent. Notably, for the VFT relation, *n* = 1, $T_{N}=T_{K}=T_{0}$.

There is an explicit straight-line behaviour of $1/I_{DO}\left(T\right)$ to the crossover $1/I_{DO}^{wEF}>1/I_{DO}^{sEF}$ $\to 1/I_{DO}^{wEF}<1/I_{DO}^{sEF}$ on cooling towards the glass temperature. The values of the coefficient *n* show that the dynamics of the squalene can be described by the critical-like Equation (4). The application of the strong electric field shifts dynamics to the VFT Equation (3) patterns.

Worth noting is the link between the activation index and the apparent fragility $m\left(T\right)=C\left(1+I_{DO}\left(T\right)\right)$, where $C=2-{log}_{10}\tau_{0}$ [56,57,58]. This suggests similar temperature dependences for $I_{DO}\left(T\right)$ and m(T). The hypothetically universal linear changes of 1/m(T) were first shown in [48] for a variety of glass formers. Notably, the same temperature dependence can be expected for 1/m(T), $1/H_{a}\left(T\right)$, and 1/(dlnτ(T)/d(1/T)) vs. *T* plots [48].

Such behaviour is shown in Figure 7. Two linear dependences intersecting at $T_{max}$ can be seen. It is worth noting that for $T>T_{max}$, the fragility index of the ultraviscous squalene is higher under the weak electric field than for the same system when the strong electric field is applied. For $T<T_{max}$, the same situation occurs.

Nonlinear dielectric studies originate from nonlinear dielectric effect investigations exhibiting changes of dielectric constant under the strong electric field. A century ago, Herweg [59] performed the first NDE test of diethyl ether, obtaining $\frac{\Delta \chi^{E}}{E^{2}}\sim 10^{-18}m^{2}V^{-2}$. Debye explained this phenomenon as the consequence of the orientation of the almost non-interacting permanent dipole moments incorporating and developing the dielectric Langevin series [60]. Two decades later, Piekara discovered a strong, positive NDE in nitrobenzene due to the dipole-dipole coupling [61]. Next, he found an anomalous positive increase in the NDE on approaching the critical consolute point in the binary mixtures of limited miscibility [62]. Not until 1999 was the latter phenomenon, as well as the similar effect in the isotropic liquid phase of rod-like liquid crystals (LC), explained as the consequence of the interactions within the pretransitional fluctuations by using the model-relation [63,64]:(14)$$\frac{\Delta \chi^{E}}{E^{2}}=\frac{\Delta \varepsilon^{E}}{E^{2}}\propto \langle \Delta M^{2}\rangle {}_{V}\chi_{T}$$
where $\langle \Delta M^{2}\rangle {}_{V}$ is for the local fluctuations of the mean square of the order parameter and $\chi_{T}$ denotes the order parameter related susceptibility (compressibility).

For the isotropic phase of LC materials $\langle \Delta M^{2}\rangle {}_{V}=const$ and $\chi_{T}\left(T\right)=\chi_{0}/\left(T-T^{*}\right)$, where *T** denotes the extrapolated temperature of a hypothetical continuous phase transition. Consequently, for rod-like LC materials, $\Delta \varepsilon^{E}/E^{2}\propto 1/\left(T-T^{*}\right)$ [55,56,57,58,59,60,61,62,63,64,65,66,67].

The pretransitional effect has been discovered for the isotropic liquid–plastic crystal phase transition in cyclo-octanol. Its form can be portrayed by the following relation [41,42,43,44,45,63,64]:(15)$$\frac{\Delta \chi^{E}}{E^{2}}\left(T\right)=e^{*}+a_{E}\left(T-T^{*}\right)+A_{E}{\left(T-T^{*}\right)}^{\phi =0.5}$$

It was indicated that the above relation could also result from Equation (14) on assuming that for ODIC-forming materials, hardly compressible semi-solid fluctuations in the liquid phase occur. Hence, $\chi_{T}\approx const$. However, the dielectric constant of pre-ODIC fluctuations differs strongly from the dielectric constant of the isotropic liquid surroundings, particularly under the strong electric field resulting from the free orientation of the permanent dipole moments. Consequently, the temperature dependence of the order parameter plays the key role in Equation (12): $\langle \Delta M^{2}\rangle {}_{V}\propto {\left(T-T^{*}\right)}^{2\beta }$, where *β* is the order parameter exponent. Assuming for the latter the classical tricritical value *β*
*=* 1/4 [39], and the linear temperature dependence for the non-critical background, one can obtain Equation (16).

Figure 8 shows the temperature changes of the NDE, exhibiting changes that can be portrayed well by Equation (15). It is worth noting that the characteristic maximum of the relation $\Delta \chi^{E}/E^{2}\left(T\right)$ occurs for the same temperature as results in Figure 5 and Figure 6. From Equation (15), the region $T_{g}<T<T_{max}$ is dominated by pretransitional previtreous fluctuations.

## 3. Experimental

This study used the broadband dielectric spectrometer (BDS by Novocontrol, Montabaur, Germany), supported by the strong electric field facility, to enable the nonlinear dielectric spectroscopy studies and the Quattro temperature control unit with a temperature stability better than ∆*T* = 0.02 K [42]. The samples were placed in a flat-parallel gold-coated measurement capacitor with plates made from Invar: a diameter of 2r=20 mm and a gap of d=0.15 mm. A quartz ring was used as the spacer. Scans of the dielectric properties were carried out in the frequency range of 0.1 Hz<f<10 MHz under the weak measuring voltage of $U_{weak}=3$ V, corresponding to the electric field $E_{weak}=0.2$ kV/cm. The scan of the dielectric properties under the strong electric field was carried out for $U_{strong}\sim 1000$ V., i.e., for $E_{strong}=60-70$ kV/cm, and for f<10 kHz, using the High Voltage facility [42,43].

To the authors’ knowledge, the nonlinear dielectric studies in glass-forming systems have mainly focused on dynamics issues and have not led to conclusive temperature dependences of the tested properties or in the functional description. This article concentrates on the static part of the NDS, the nonlinear dielectric effect (NDE) [43,44,45], which has, to date, rarely been tested for glass-forming systems: $\varepsilon \left(E\right)=\varepsilon \left(E\to 0\right)+\Delta \varepsilon E^{2}+\ldots$, where ε(E→0)=ε represents the dielectric constant, and then:(16)$$NDE:=\frac{\Delta \varepsilon }{E^{2}}=\frac{\varepsilon \left(E\right)-\varepsilon }{E^{2}}=\frac{\Delta \chi }{E^{2}}$$
where *χ* denotes the dielectric susceptibility: χ=ε−1.

The notable features for the measurement under the strong electric field are the bulk gap of the measurement capacitor and the degassing of samples immediately prior to the experiment. These essentially reduce the risk of gas bubbles in the sample, which can qualitatively distort measurement results under the strong electric field. Notable was the extremely low electric conductivity, which caused no heating of the sample during measurements under the strong electric field, as was a well-evidenced basing factor in related studies. Finally, the occurrence of the condition $NDE\propto E^{2}$ has been carefully tested. The key targets of the dielectric studies under the strong electric field were to locate the temperature changes of the tested properties, which are possible for functional evaluations.

## 4. Conclusions

This article presents evidence for the long-range pretransitional/previtreous behaviours of two static properties: the dielectric constant and its strong electric field-related counterpart, the nonlinear dielectric effect. The possibility of their functional characterisations by Equation (7), with the ‘Mossotti Catastrophe’ form, and Equation (13) resembling the formula developed for critical liquids is notable. The analysis of the dynamic properties based on the activation energy index excluded the VFT relation (associated with the coefficient *n* = 1) as a validated tool for portraying the evolution of the primary relaxation time in squalene, both under weak and strong electric fields. These results question the commonly applied ‘Stickel operator’ routine, used as the tool for determining the dynamic crossover between the high- and low-temperatures dynamic domains in the supercooled state. The strong electric field radically affects the distribution of relaxation times, the form of the evolution of the primary relaxation time, and the fragility metric.

## Figures and Tables

**Figure 1 molecules-26-05811-f001:**
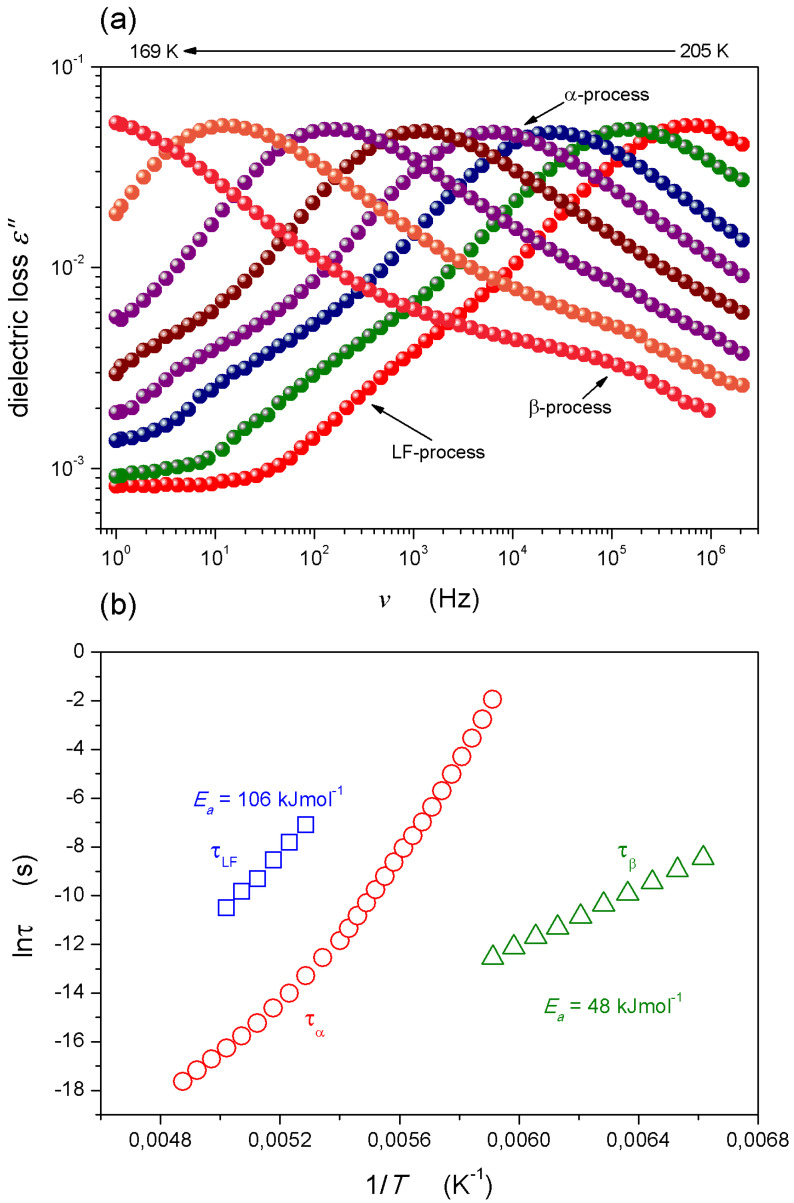
(**a**) The broadband dielectric loss spectra for the supercooled squalene at temperatures of 201–169 K; (**b**) The distribution of relaxation times: the Arrhenius behaviour for low-frequency and secondary processes, and the VFT behaviour for the primary relaxation. The activation energies were calculated with the Arrhenius equation $\tau \left(T\right)=\tau_{0}exp\left(\frac{E_{a}}{RT}\right)$. For obtaining the β-process, the temperature range of 201–151 K was used.

**Figure 2 molecules-26-05811-f002:**
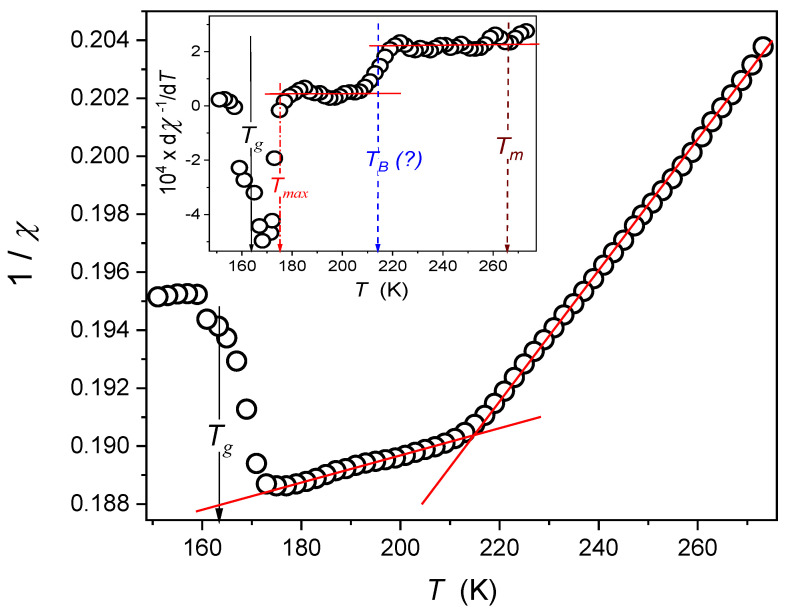
The temperature evolution of the reciprocal of the dielectric constant in supercooling squalene. Red, solid lines are related to the description via Equation (7). The inset is for the distortions-sensitive derivative of the results from the central part of the plot. It additionally confirms the description via Equation (7) by revealing some distortions close to *T_g_* and strong changes near the hypothetical dynamic crossover temperature $T_{B}=215\;K\pm 5$.

**Figure 3 molecules-26-05811-f003:**
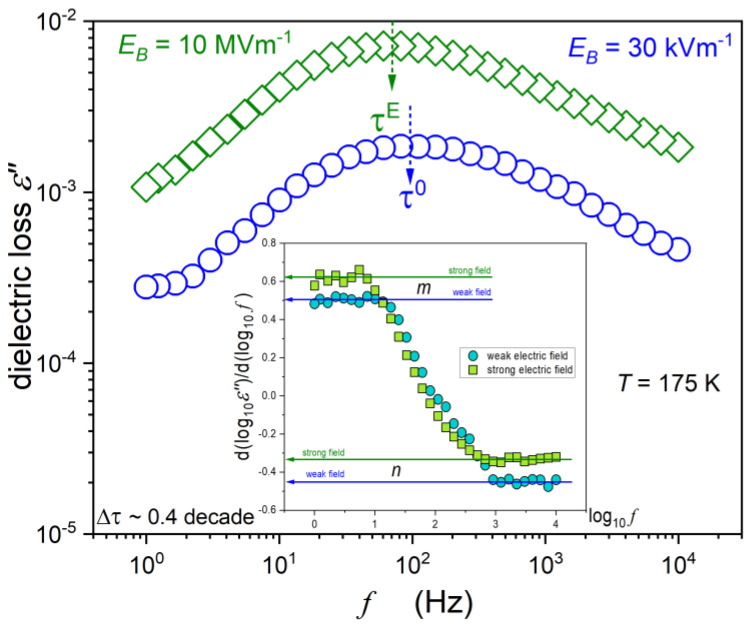
The dielectric loss spectra for low- and high-voltage applied to the samples. The inset shows the derivatives of experimental data from the main part of the plot and recalls the Jonsher’s analysis via Equation (9).

**Figure 4 molecules-26-05811-f004:**
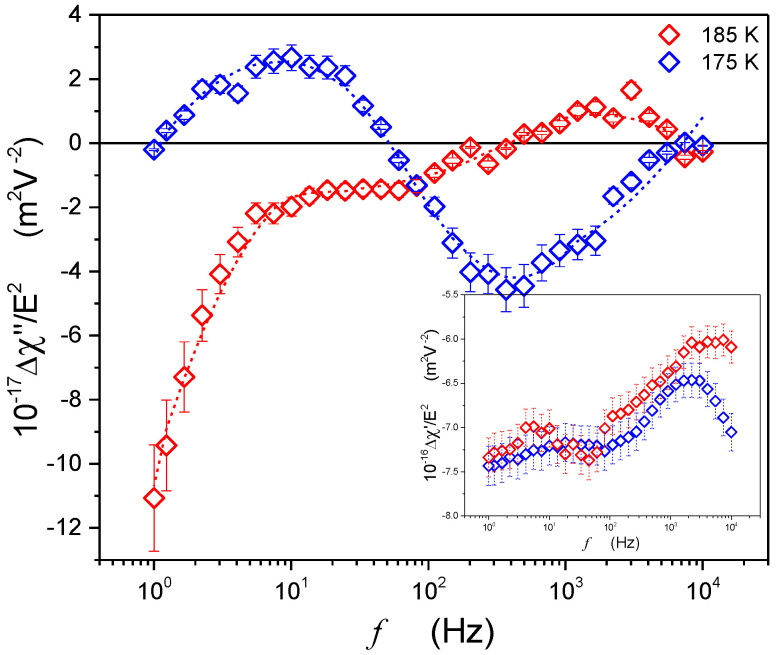
Examples of nonlinear dielectric spectra for the imaginary part of the dielectric susceptibility $\Delta \varepsilon^{\prime \prime }\equiv \Delta \chi^{\prime \prime }=\chi_{hi}^{\prime \prime }-\chi_{lo}^{\prime \prime }$, where χ=ε−1, in the supercooled domain near the glass transition for $T<T_{B}^{*}$ and $T>T_{B}^{*}$. The insert shows the real part $\Delta \varepsilon^{\prime }\equiv \Delta \chi^{\prime }=\chi_{hi}^{\prime }-\chi_{lo}^{\prime }$.

**Figure 5 molecules-26-05811-f005:**
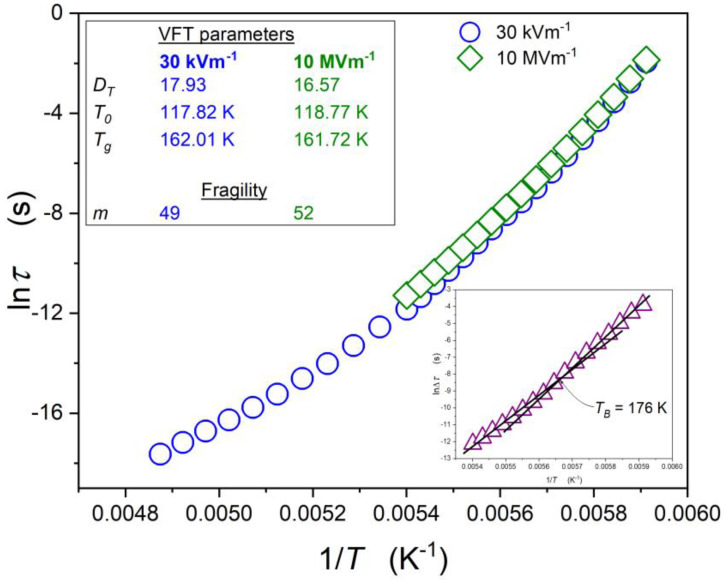
A relaxation map for the supercooled squalene in the weak (30 kV/m) and strong (10 MV/m) electric fields. The insert presents a difference between relaxation times $\Delta \tau =\tau^{hi}-\tau^{lo}$, with the visible singularity *T_B_ = T_max_* = 176 K (compare with Figure 2).

**Figure 6 molecules-26-05811-f006:**
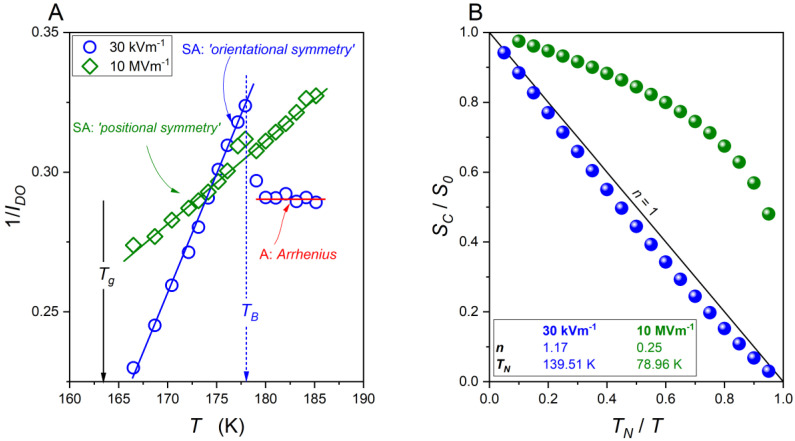
(**A**) The changes in the reciprocal of the apparent activation index *I_DO_* (Equation (10)) for supercooled squalene, based on experimental data given in Figure 5. (**B**) Configurational entropy *S_C_* for squalene under the weak (blue) and strong (green) electric fields calculated from Equation (13). Parameters with *n* = 1.17, *T_N_* = 139.51 K, and *n* = 0.25; *T_N_* = 78.96 K for weak and strong electric fields, respectively, were obtained using Equation (11).

**Figure 7 molecules-26-05811-f007:**
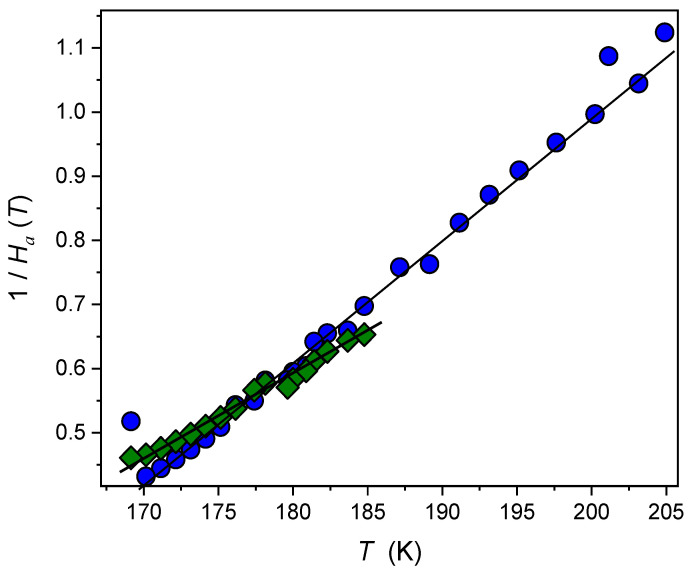
The linear temperature changes of the reciprocal of the apparent activation enthalpy in supercooled squalene under weak and strong electric fields when approaching the glass temperature. Note the relation to the apparent fragility: $1/H_{a}\left(T\right)\propto 1/m\left(T\right)$.

**Figure 8 molecules-26-05811-f008:**
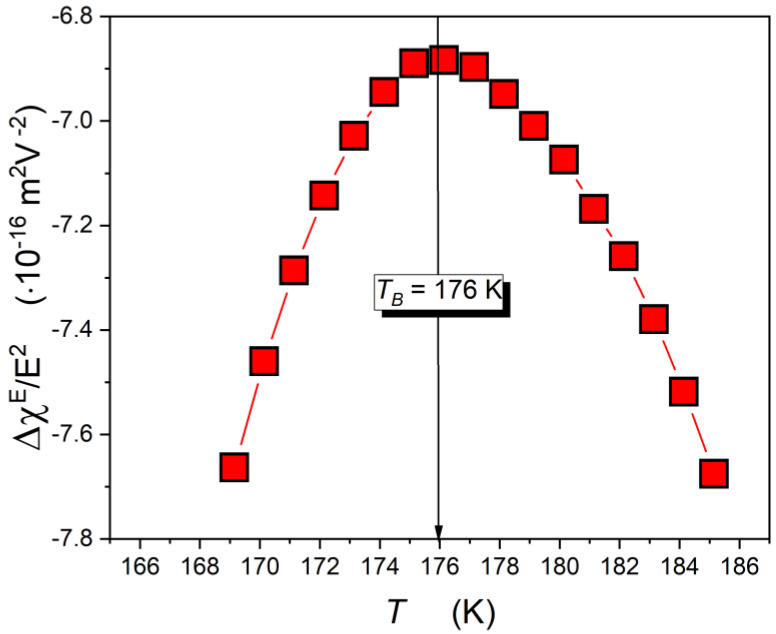
The nonlinear dielectric effect (NDE) on approaching the glass temperature in ultraviscous, supercooled squalene. The curve portraying experimental data is related to Equation (15).

## Data Availability

The data supporting the findings of this study are available within the article.

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
