# Peer review of "Long-Range Static and Dynamic Previtreous Effects in Supercooled Squalene—Impact of Strong Electric Field"

_molecules, 2021, doi:10.3390/molecules26195811_

Round 1
Reviewer 1 Report
Please see attachment
Author Response
Dear Reviewer,
I am thankful for all comments and suggestions. Please find explanations below.
Ad 1.
We have modified cited statement.
Ad 2.
Correct relationship has been introduced.
Ad 3.
The red balls correspond to 205 K and the pink ones to 169 K. The temperature is decreasing from the right to the left.
Ad 4.
TB is defined as a cross-over temperature between ergodic and non-ergodic regions. It is true that Fig. 2 presents a failure of Eq. (8) which corresponds to Mossotti catastrophe.
Ad 5.
We add error bars. Pease remember that this method differs from the others presented in existed evidence. Measurements were carried out in static and relaxation regions as well. The latter has a huge impact on a shape of Δχ.
Ad 6.
Fig. 5 present is the relaxation map showing VFT parameters though we showed the inadequacy of such a description, due to present mainstream point of view and data representation.
Ad 7.
We do not want to promote ourselves. 1/3 of all cited publications belongs to us, which is not a deviation from the norm among other research groups. We believe that all citations used in this paper are adequate.
Regards,
Szymon Starzonek
Reviewer 2 Report
The paper presents the evidence for long-range pre-vitreous changes of dielectric constant associated with specific pre-vitreous evolution of dynamic properties of supercooled squalene. The paper is of good quality and shall be published with a couple of amendments to account for the comments given below.
The comments are as follows:
Lines 64-65: Authors shall be aware that the Arrhenius-type behaviour of viscosity and hence of the relaxation times at low and high temperatures is a known experimental result analysed in detail for glass forming liquids and ancient glasses and denying low-temperature diverging predictions of other models, see e.g. the following publications:
- R.H. Doremus. Viscosity of silica. J. Appl. Phys., 92, 7619–7629(2002).
- J. Zhao et al. Using 20-million-year-old amber to test the super-Arrhenius behaviour of glass-forming systems. Nat. Commun. 2013, 4, 1783 (2013).
- G.B. McKenna, J. Zhao, J. Accumulating evidence for non-diverging time-scales in glass-forming fluids. J. Non-Cryst. Solids, 407, 3–13 (2015).
- H. Yoon, G.B. McKenna. Testing the paradigm of an ideal glass transition: Dynamics of an ultrastable polymeric glass. Sci. Adv., 4, eaau5423 (2018).
- D.S. Sanditov, M.I. Ojovan. Relaxation aspects of the liquid—glass transition. Physics Uspekhi 62 (2) 111 - 130 (2019). https://doi.org/10.3367/UFNr.2018.04.038319
- M.I. Ojovan. On viscous flow in glass-forming organic liquids. Molecules, 25 (17), 4029, 13 p. (2020). doi:10.3390/molecules25174029
- J. Deubener. Viscosity of glass-forming melts. Chapter 4.1 in Encyclopedia of Glass Science, Technology, History, and Culture. Wiley (2021). https://doi.org/10.1002/9781118801017.ch4.1
Lines 80-81: the above-noted work [5] gives many relationships evidencing on non-singularities.
Line 263: The sentence “The application of Eq. (10) f requires a priori knowledge of the apparent activation…” seems to contain misprints (e.g. f) and shall refer to Eq. (12) instead.
Line 266: see the above Reference [5] for the characteristic timescales along with their interpretation for many glass forming systems.
Line 270 and those starting from line 346 contain some formatting issues, e.g. the text cannot belong to the figure caption.
Author Response
Dear Reviewer,
I am thankful for your comments. All suggestions have been taken into account and placed into text.
Best regards,
Szymon Starzonek
Reviewer 3 Report
This is a very interesting and well written paper exploring new evidences concerning the changes in both dynamic and static properties of squalene upon cooling from above Tg. References to the prior works were cited and discussed adequately and the conclusions were postulated in logical and reasonable fashion. This reviewer would recommend publishing it after minor revision addressing the following:
- Line 36, delete duplicated “Arrhenius dependence”
- Line 64, missing subscript on D subT
- Line 142, “were find” to “were to find”
- Line 165, the last name of Prof. Wolynes was misspelled with extra s.
- Line 186, “gas” to “gap”
- Line 191-192, check epsilon -1 equals 1 over a+bT, is it correct?
- Line 192, define Tb
- Line 193, is a=0.181 instead of 1 being on subscript?
- Figure 2, define Tmax.
- Line 259, “can offer” to “we can offer is”
- In later part of the paper, equation number mentioned in text or figure seemed to be off by 2. Please check all to make sure. Below are what this reviewer found:
Figure 6 caption, Line 289-293 334 340 342 345
Author Response
Dear Reviewer,
I am thankful for your comments. All corrections suggested by Reviewer are placed into text.
Best regards,
Szymon Starzonek
Round 2
Reviewer 1 Report
See the attached file. It would be nice to see the author's response as a separate file in future manuscripts.

Author Response
Attached file includes the same comments I responded before. Please find manuscript with corrections.
Regards,
Szymon Starzonek